# Enhancing the Ranking Context of Dense Retrieval through Reciprocal Nearest Neighbors

**George Zerveas**
Brown University, USA
george_zerveas@brown.edu

**Navid Rekabsaz**
JKU Linz
LIT AI Lab, Austria
navid.rekabsaz@jku.at

**Carsten Eickhoff**
University of Tübingen, Germany
c.eickhoff@acm.org

## Abstract

Sparse annotation poses persistent challenges to training dense retrieval models, for example by distorting the training signal when unlabeled relevant documents are used spuriously as negatives in contrastive learning. To alleviate this problem, we introduce evidence-based label smoothing, a novel, computationally efficient method that prevents penalizing the model for assigning high relevance to false negatives. To compute the target relevance distribution over candidate documents within the ranking context of a given query, those candidates most similar to the ground truth are assigned a non-zero relevance probability based on the degree of their similarity to the ground-truth document(s). To estimate relevance we leverage an improved similarity metric based on reciprocal nearest neighbors, which can also be used independently to rerank candidates in post-processing. Through extensive experiments on two large-scale ad hoc text retrieval datasets, we demonstrate that reciprocal nearest neighbors can improve the ranking effectiveness of dense retrieval models, both when used for label smoothing, as well as for reranking. This indicates that by considering relationships between documents and queries beyond simple geometric distance we can effectively enhance the ranking context.[1]

## 1 Introduction

The training of state-of-the-art ad-hoc text retrieval models (Nogueira and Cho, 2020; Santhanam et al., 2021; Zhan et al., 2021; Ren et al., 2021b,a; Gao and Callan, 2021; Zhang et al., 2022; Lu et al., 2022), which are based on transformer Language Models, relies on large-scale datasets that are sparsely annotated, typically comprising only a small number of relevance judgements for each query.[2] These labels are usually derived from sub-

mitting the strongest pseudo-relevance signals in user click logs to human judges for verification. Despite potential future endeavors to extend annotation, this sparsity and the resulting issue of false negatives (Qu et al., 2021; Zhou et al., 2022) – i.e., only a minuscule fraction of all documents pertinent to a query are ever seen by users or judges and identified as relevant – will inevitably persist. To eliminate the sparsity, it would be necessary to acquire either human judgements, or perhaps expensive evaluations from Large Language Models, to verify the relevance of the *entire* document collection (typically tens of millions of documents) with respect to *every* query in the dataset, leading to an intractable Cartesian product. Consequently, it is crucial to explore optimizing the utilization of existing information, and extract richer structural relationships between documents and queries, without additional annotations.

To this end, in the present work we follow a two-pronged approach: first, we employ the concept of *reciprocal nearest neighbors* (rNN) to improve the estimation of semantic similarity between embeddings of queries and documents. Two documents $c_i$ and $c_j$ are said to be $k$-reciprocal nearest neighbors if $c_j$ is within the $k$-nearest neighbors of $c_i$, and at the same time $c_i$ is within the $k$-nearest neighbors of $c_j$. Second, we attempt to enhance the query-specific *ranking context* used to train dense retrievers, going beyond the notion of using mined candidates merely as negatives for contrastive learning. By ranking context we mean a set of documents that are in meaningful relationship to the query and are *jointly* evaluated with respect to their relevance to the query (Ai et al., 2018; Zerveas et al., 2022). Specifically, we use the similarity of ground-truth documents to candidates in the same ranking context as the query as evidence to guide the model's predicted relevance probability distribution over candidates.

Dense retrieval, the state-of-the-art approach for

---

[1] Our code and other resources are available at:
https://github.com/gzerveas/CODER

[2] E.g., on average 1.06 documents per query in MS MARCO, Bajaj et al., 2018.

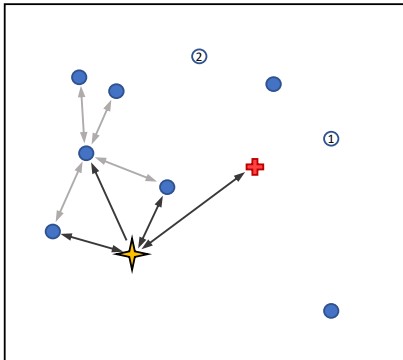

Figure 1: Query (yellow star), positive (red cross) and negative (full blue circles) document embedding vectors in a shared 2D representation space. Based on top-4 Nearest Neighbors, the positive would be ranked lower than the 3 nearest neighbors of the query. When using top-4 Reciprocal Nearest Neighbors, its ranking is improved, because of its reciprocal relationship to the query, which one of 3 nearest neighbors of the query lacks. Adding an extra negative to the context (circle #1) does not affect this ranking, but the second extra negative (#2) disrupts the reciprocal relationship, becoming the 4th nearest neighbor of the positive.

single-stage ad-hoc retrieval, is premised on modeling relevance between a query and a document as the geometric proximity (e.g., dot-product or cosine similarity) between their respective embeddings in the common representation vector space. Top retrieval results are therefore the documents whose embeddings are the nearest neighbors of the query embedding. However, this modeling assumption may be sub-optimal: previous work in the field of image re-identification has shown that, while geometric similarity can easily differentiate between candidate embeddings in near proximity from a query embedding, the differences between relevance scores of candidate embeddings become vanishingly small as distance from the query increases (Qin et al., 2011). It was found that the degree of overlap between sets of reciprocal nearest neighbors can be used to compute an improved measure of similarity between query and candidate embeddings (Zhong et al., 2017).

Moreover, geometric similarity is used in mining "hard" negatives, which have been consistently found to improve performance compared to random in-batch negatives (Xiong et al., 2020; Zhan et al., 2021; Qu et al., 2021; Zerveas et al., 2022). Hard negatives are typically the top-ranked candidates retrieved by a dense retriever (nearest neighbors to a query embedding) that are not explicitly annotated

as relevant in the dataset.

On the one hand, the effectiveness of mined negatives is limited by how effectively this dense retriever can already embed queries and relevant documents in close proximity within the shared representation space, although the periodical or dynamic retrieval of negatives during training can partially alleviate this problem (Xiong et al., 2020; Zhan et al., 2021). On the other hand, when the retriever used to mine hard negatives indeed succeeds in retrieving candidates that are semantically relevant to the query, these are often not marked as positives due to the sparsity of annotation and are thus spuriously used as negatives for contrastive learning (false negatives)[3], confounding the training signal (Qu et al., 2021; Zhou et al., 2022).

For this reason, in this work we investigate to what degree these issues can be mitigated through the use of reciprocal nearest neighbors, essentially extracting additional relationship information between queries and documents beyond flat geometric distances, such as the local degree of node connectivity. Furthermore, unlike all existing dense retrieval methods, instead of using candidates exclusively as negatives, we propose using their estimated similarity to the ground-truth document(s) as evidence for label smoothing; we thus redistribute probability weight in the target score distribution from the ground truth to a larger number of likely false negatives.

Finally, our work places a strong emphasis on computational efficiency: label smoothing can be performed entirely offline on CPUs and can be trivially parallelized, while no latency is introduced during training and our models can be trained (e.g., on MS MARCO) within hours, using a single GPU with a batch size of 32. Reranking based on reciprocal nearest neighbors, when used, introduces a few milliseconds latency per query on a CPU.

By contrast, the current state-of-the-art dense retrieval methods (e.g. (Qu et al., 2021; Ren et al., 2021b)) depend on the existence of better performing, but computationally demanding re-ranking models such as cross-encoders, which are typically run offline on several GPUs with huge batch sizes and are used either for pseudo-labeling additional training data, for discarding negatives which are likely unlabeled positives (i.e., false negatives), or directly for distillation through a teacher-student

---

[3]Qu et al., 2021 estimate that about 70% of the top 5 candidates retrieved by a top-performing dense retrieval model that are not labeled as positive are actually relevant.

training scheme. However, besides the very high computational cost of such pipelines, the existence of a model that is more powerful than the retrieval model we wish to train is a very restrictive constraint, and cannot be taken for granted in many practical settings.

Our main contributions are:
(1) We propose evidence-based label smoothing, a novel method which mitigates the problem of false negatives by leveraging the similarity of candidate documents *within the ranking context of a query* to the annotated ground truth in order to compute soft relevance labels. Different from existing methods like teacher-student distillation or pseudo-labeling, our approach does not rely on the existence of more powerful retrieval methods.
(2) We explore the applicability of the concept of reciprocal nearest neighbors in improving the similarity metric between query and document embeddings in the novel setting of ad-hoc text retrieval.
(3) Through extensive experiments on two different large-scale ad-hoc retrieval datasets, we demonstrate that the concept of reciprocal nearest neighbors can indeed enhance the ranking context in a computationally efficient way, both when reranking candidates at inference time, as well as when applied for evidence-based label smoothing intended for training.

## 2 Related work

Our proposed label smoothing, which encourages the model to assign higher relevance scores to documents intimately related to the ground truth, conceptually finds support in prior work that proposed *local relevance score regularization* (Diaz, 2007), adjusting retrieval scores to respect local inter-document consistency. Despite the entirely different methodology, both methods are premised on the intuition that documents lying closely together in the representation vector space should have similar scores; this in turn is related to the *cluster hypothesis*, which states that closely related documents (and thus proximal in terms of vector representations) tend to be relevant to the same request (Jardine and van Rijsbergen, 1971).

Zerveas et al., 2022 recently argued that jointly scoring a large number of candidate documents (positives and negatives) closely related to the same query within a list-wise loss constitutes a query-specific ranking context that benefits the assessment of relevance of each individual candidate doc-

ument with respect to the query. Thus, they extended well-established insights and empirical findings from Learning-to-Rank literature (Cao et al., 2007; Ai et al., 2019, 2018) to the realm of dense retrieval through transformer-based Language Models. While in-depth annotation of candidate documents (i.e., hundreds of relevance judgements per query) explicitly provides a rich context for each query in Learning-to-Rank datasets (Qin et al., 2010; Chapelle and Chang, 2010; Dato et al., 2016), such information is not available in the sparsely annotated, large-scale datasets used to train dense retrieval models. The relationship exploited thus far to "build a context" (practically, this means mining hard negatives), is simply that of geometric proximity between the embeddings of a query and candidate documents.

Addressing the problem of sparse annotation, several works have utilized the relevance estimates from supervised (e.g. Hofstätter et al., 2021; Qu et al., 2021; Ren et al., 2021b) or unsupervised (e.g. lexical: Dehghani et al., 2017; Haddad and Ghosh, 2019) retrieval methods or other dataset-specific heuristics (e.g. bibliography citations: Moro and Valgimigli, 2021) to derive soft labels for documents used to train a model, e.g., in a teacher-student distillation scheme. In this work, we instead shift the perspective from assigning labels based on similarity *with respect to the query*, to similarity *with respect to the ground-truth document(s)*, but within a query-specific ranking context. We furthermore leverage the concept of reciprocal nearest neighbors, introduced as a reranking method for image re-identification (Qin et al., 2011; Zhong et al., 2017), to improve the similarity estimate.

False negatives have been identified as a significant challenge by prior work, which has employed powerful but computationally expensive cross-encoders (Nogueira and Cho, 2020) to discard documents that receive a high similarity score to the query and are thus likely relevant from the pool of hard negatives (Qu et al., 2021; Ren et al., 2021b). However, discarding top-ranking hard negatives also discards potentially useful information for training.

Recently, Zhou et al. (2022) tackled the problem of false negatives through selective sampling of negatives around the rank of the ground-truth document, avoiding candidates that are ranked either much higher than the ground truth (probable false negatives) or much lower (too easy negatives). This

approach differs from ours in the perspective of similarity (query-centric vs ground-truth-centric), and in the fact that information is again discarded from the context, as only a small number of negatives is sampled around the positive. Additionally, a query latency of up to 650 ms is added during training.

Ren et al. (2021a) leverage the similarity of candidate documents to the ground truth document (positive), but in a different way and to a different end compared to our work: all documents in the batch ("in-batch negatives") as well as retrieved candidates are used as negatives in an InfoNCE loss term, which penalizes the model when it assigns a low similarity score between a single positive and the query compared to the similarity score it assigns to pairs of this positive with all other candidates. Thus, it requires that the ground truth lies closer to the query than other candidates, but the detrimental effect of false negatives on the training signal fully persists.

By contrast, our method jointly takes into account all positives and other candidates in the ranking context, and through a KL-divergence loss term requires that the predicted relevance of the query with respect to all documents in the ranking context has a similar probability distribution to the target distribution, i.e., the distribution of similarity between all ground truth positives and all candidate documents in the context. False negatives are thus highly likely to receive a non-zero probability in the target distribution, and the penalty when assigning a non-zero relevance score to false negatives is lower.

## 3 Methods

### 3.1 Similarity metric based on Reciprocal Nearest Neighbors

Nearest Neighbors are conventionally retrieved based on the geometric similarity (here, inner product) between embedding vectors of a query $q$ and candidate document $c_i$: $s(q, c_i) = \langle x_q, x_{c_i} \rangle$, with $x_q = m(q)$ and $x_{c_i} = m(c_i)$ embeddings obtained by a trained retrieval model $m$. We can additionally define the Jaccard similarity $s_J$ that measures the overlap between the sets of *reciprocal neighbors* of $q$ and $c_i$. We provide a detailed derivation of $s_J$ in Appendix A.1.

Instead of the pure Jaccard similarity $s_J$, we use a linear mixture with the geometric similarity $s$ controlled by hyperparameter $\lambda \in [0, 1]$:

$$s^*(q, c_i) = \lambda\, s(q, c_i) + (1 - \lambda)\, s_J(q, c_i), \quad (1)$$

which we found to perform better both for reranking (as in Zhong et al., 2017), as well as for label smoothing.

Importantly, unlike prior work (Qin et al., 2011; Zhong et al., 2017), which considered the entire gallery (collection) of images as a *reranking context* for each probe, we only use as a context a limited number of candidates previously retrieved for each query. This is done both for computational tractability, as well as to constrain the context to be query-specific when computing the similarity of documents to the ground truth; documents can otherwise be similar to each other with respect to many different topics unrelated to the query. We empirically validate this choice in Section 5.1.

### 3.2 Evidence-based label smoothing

*Uniform* label smoothing is a well-established technique (Goodfellow et al., 2016) that is used to mitigate the effects of label noise and improve score calibration, and was recently also employed for contrastive learning (Alayrac et al., 2022). It involves removing a small proportion $\epsilon \in [0, 1]$ of the probability mass corresponding to the ground-truth class and uniformly redistributing it among the rest of the classes, thus converting, e.g., a 1-hot vector $\mathbf{y} = [1, 0, \ldots, 0] \in \mathbb{R}^N$ to:

$$\mathbf{y}^* = [1-\epsilon,\ \epsilon/(N-1), \ldots, \epsilon/(N-1)] \in \mathbb{R}^N \quad (2)$$

Nevertheless, naively distributing the probability mass $\epsilon$ uniformly among all candidates, as in Eq. (2), would result in true negatives predominantly receiving a portion of it, apart from the small number of false negatives[4].

For this reason, we instead propose correcting the sparse annotation vectors by selectively distributing relevance probability among negatives that are highly likely to be positive, or at least are ambiguous with respect to their relevance to the query. The proportion of probability mass each candidate shall receive depends on its degree of similarity to the annotated ground-truth document, which can be quantified by the Jaccard distance of Eq. (11), if we wish to exclusively consider reciprocal nearest neighbors, or the mixed geometric-Jaccard distance of Eq. (1), which allows any candidate close to the ground-truth to be considered.

---

[4]Indeed, Qu et al. (2021) observe that among mined "hard negative" candidates, the percentage of false negatives falls to 4% by rank 40.

**Algorithm 1** Evidence-based label smoothing

---

**Require:** Dense retrieval model $m$, set of queries $\mathcal{Q}$, document collection $\mathcal{C}$, set of all ground-truth label documents per query $\bigcup \mathcal{L}(q)$, $\forall q \in \mathcal{Q}$

1: Compute embedding vectors $x_q = m(q)$, $\forall q \in \mathcal{Q}$ and $x_{c_i} = m(c_i)$, $\forall c_i \in \mathcal{C}$.

2: **for** each query $q$ **do**

3:      Retrieve top-$N$ Nearest Neighbors per query based on geometric similarity: $s(q, c_i) = \langle x_q, x_{c_i} \rangle$ for all $c_i \in C$.

4:      **for** each candidate $c_i$, $i = 1, \dots, N$ **do**

5:          Compute relevance score $r''$ as mixed geometric and reciprocal-NN Jaccard similarity $s_J$ with respect to all ground-truth documents $l$:

$$r''(q, c_i) = \frac{1}{|\mathcal{L}(q)|} \sum_{l \in \mathcal{L}(q)} s^*(l, c_i),$$

$$s^*(l, c_i) = \lambda \cdot s(l, c_i) + (1 - \lambda) \cdot s_J(l, c_i),$$
$$0 < \lambda < 1$$

6:          Transform scores by applying normalization function $f_n$, boost factor $b$ and cut-off threshold $n_{\max}$:

$$r'(q, c_i) = \begin{cases} b \cdot f_n\left(r''(q, c_i)\right) & \text{if } c_i \in \mathcal{L}(q), \\ -\infty & \text{if } i > n_{\max}, \\ f_n\left(r''(q, c_i)\right) & \text{otherwise.} \end{cases}$$

7:      **end for**

8: **end for**

9: Fine-tune model $m$ with target distribution: $\mathbf{r}(q) = \text{softmax}\left(\mathbf{r}'(q)\right)$, and loss function:
$\mathfrak{L}(\mathbf{r}(q), \hat{\mathbf{s}}(\mathbf{q})) = D_{\text{KL}}(\mathbf{r}(q) \,||\, \hat{\mathbf{s}}(q))$,
where $\hat{\mathbf{s}}(q) = \text{softmax}(\hat{\mathbf{s}}'(q)/T)$ is the model-predicted score distribution, with $T$ a learnable temp. param.

---

Since the value range of similarity scores that each model outputs is effectively arbitrary, before applying a softmax to obtain a distribution over candidates, we (1) perform transformations (e.g., max-min or std-based, see Appendix A.5.1) and multiply the values of the original ground-truth documents by a factor $b > 1$ to normalize the range and increase the contrast between the top and trailing candidates, and (2) we limit the number of candidates that receive a probability above 0 to the top $n_{\max}$ candidates in terms of their similarity to the ground-truth document(s). We found that these transformations primarily depend on the dataset rather than the model, and that training without limiting $n_{\max}$ leads to overly diffuse score distributions. In case more than one ground-truth documents exist for the same query, the similarity of each candidate is the mean similarity over all ground-truth documents (see Algorithm 1).

### 3.3 Computational efficiency

Computing rNN similarity involves computing pairwise similarities among $N + 1$ ranking context elements (including the query), and reranking re-

quires sorting the $N$ candidates by their final similarity. The computational cost is thus $O(N^2)$ and $O(N \log N)$, respectively; if we are only interested in the top-$k$ reranked candidates, the latter can be reduced to $O(N \log k)$. We find (Sections 5.1, A.4) that a small subset of the full ranking context with size $N_r < N$ is generally sufficient when computing rNN-based similarities. For MS MARCO, $N_r = 60$ and the delay per query when reranking on a single CPU and core (AMD EPYC 7532, 2400 MHz) is about 5 ms (Fig. 7).

Evidence-based label smoothing imposes no cost during training or inference; it only requires *offline* computation of rNN-similarities for each query context $N_r$ and sorting/top-$k$ as above, followed by simple vectorized transformations, e.g. max-min normalization. Furthermore, all computations above can be trivially ('*embarrassingly*') parallelized in a multi-CPU/core setup.

## 4 Experimental setting

**Datasets.** To evaluate the effectiveness of our methods, we use two large-scale, publicly available ad-hoc retrieval collections: the MS MARCO Passage Retrieval dataset (Bajaj et al., 2018), and TripClick, a health document retrieval dataset (Rekabsaz et al., 2021b). Each has distinct characteristics and represents one of the two realistic data settings practically available for training dense retrieval models (see details in Appendix A.2, A.3).

**Baselines.** To compute the similarity metric based on reciprocal nearest neighbors, and thus the scores used to either rerank candidates at inference time or calculate the smoothed labels for training, we only need access to the encoder extracting the document and query embeddings. The methods we propose are therefore applicable in principle to any dual-encoder dense retriever. However, we eschew training pipelines based on cross-encoders, both to ensure computational efficiency, as well as to eliminate the dependence on more powerful retrieval methods. Instead, we choose CODER (Zerveas et al., 2022), a fine-tuning framework that enhances the performance of dense retrievers used as "base models" through a large ranking context of query-specific candidate documents and a list-wise loss: it serves as a natural framework to evaluate evidence-based label smoothing, because it allows us to directly utilize a large number of soft labels per query, while being very light-weight computationally.

Following Zerveas et al. (2022), we select the

| Model | MS MARCO dev.small | | | MS MARCO dev | | | TREC DL 2019 | | | TREC DL 2020 | | |
|---|---|---|---|---|---|---|---|---|---|---|---|---|
| | MRR | nDCG | Recall | MRR | nDCG | Recall | MRR | nDCG | Recall | MRR | nDCG | Recall |
| RocketQAv2 (Ren et al., 2021b) | 0.388 | - | - | - | - | - | - | - | - | - | - | - |
| ERNIE-Search (Lu et al., 2022) | 0.401 | - | - | - | - | - | - | - | - | - | - | - |
| AR2 (Zhang et al., 2022) | 0.395 | - | - | - | - | - | - | - | - | - | - | - |
| AR2 + SimANS (Zhou et al., 2022) | 0.409 | - | - | - | - | - | - | - | - | - | - | - |
| TAS-B | 0.344 | 0.408 | 0.619 | 0.344 | 0.407 | 0.618 | 0.875 | 0.659 | 0.222 | **0.832** | 0.620 | 0.302 |
| R. TAS-B | **0.347** | **0.411** | **0.625** | **0.346** | **0.410** | **0.623** | **0.886** | **0.664** | **0.226** | 0.828 | **0.627** | **0.311** |
| CODER(TAS-B) | 0.355 | 0.419 | 0.633 | 0.353 | 0.416 | 0.627 | **0.857** | 0.668 | 0.224 | 0.844 | 0.623 | 0.306 |
| R. CODER(TAS-B) | **0.357** | **0.421** | **0.637** | **0.354** | **0.418** | **0.631** | 0.853 | **0.679** | **0.231** | **0.860** | **0.634** | **0.317** |
| CoCondenser | 0.381 | 0.446 | 0.665 | **0.381** | 0.446 | 0.664 | **0.879** | 0.656 | **0.226** | **0.833** | 0.618 | 0.301 |
| R. CoCondenser | **0.384** | **0.449** | **0.670** | **0.381** | **0.447** | **0.666** | 0.877 | **0.658** | **0.226** | **0.833** | **0.627** | **0.306** |
| CODER(CoCond) | 0.382 | 0.447 | 0.668 | 0.382 | 0.447 | 0.665 | **0.895** | 0.655 | 0.228 | **0.844** | 0.639 | 0.314 |
| R. CODER(CoCond) | **0.384** | **0.450** | **0.671** | **0.383** | **0.448** | **0.667** | **0.895** | **0.664** | **0.230** | **0.844** | **0.641** | **0.316** |

Table 1: Recip. NN reranking, MS MARCO collection. Metrics cut-off @10. **Bold**: best in model class. As a reference, at the top we include all SOTA dense retrieval models from literature that ourperform the methods we evaluated, noting that, unlike ours, they all rely heavily on cross-encoders for training (e.g. distillation, ranking, pseudolabeling etc). Blue: our contributions.

| Model | DCTR Head | | RAW Head | | |
|---|---|---|---|---|---|
| | MRR | nDCG | MRR | nDCG | Recall |
| BM25[1] | 0.276 | 0.224 | - | 0.199 | 0.128 |
| BERT-Dot (SciBERT)[2] | 0.530 | 0.243 | - | - | - |
| BERT-Cat (SciBERT)[2] | 0.595 | 0.294 | - | - | - |
| RepBERT [abbrev: RB] | 0.526 | 0.255 | 0.574 | 0.344 | 0.199 |
| R. RepBERT | 0.525 | 0.256 | 0.575 | 0.346 | 0.200 |
| CODER(RB) | 0.634 | 0.316 | 0.674 | 0.419 | 0.234 |
| R. CODER(RB) | 0.638 | 0.317 | 0.679 | 0.418 | 0.234 |
| RB + CODER(RB) | 0.637 | 0.318 | 0.679 | 0.421 | 0.235 |
| RB + R. CODER(RB) | **0.641** | **0.319** | **0.681** | **0.422** | **0.236** |

Table 2: Recip. NN reranking, TripClick Test (cut-off @10). **Bold**: overall best, underline: best in model class. Row [1]: from (Rekabsaz et al., 2021b), [2]: (Hofstätter et al., 2022). Blue: our contributions.

following base models subjected to CODER fine-tuning :

1. RepBERT (Zhan et al., 2020), a BERT-based model with a typical dual encoder architecture which underpins all state-of-the-art dense retrieval methods, trained using a triplet Max-Margin loss.
2. TAS-B (Hofstätter et al., 2021), one of the top-performing dense retrieval methods on the MS MARCO / TREC-DL 2019, 2020 datasets, which has been optimized with respect to their training process, involving a sophisticated selection of negative documents through clustering of topically related queries.
3. CoCondenser (Gao and Callan, 2021), the state-of-the-art dense retrieval model, *excluding* those which make use of heavyweight cross-encoder (query-document term interaction) teacher models or additional pseudo-labeled data samples; it relies on corpus-specific, self-supervised pre-training through a special architecture and contrastive loss component.

## 5 Results and Discussion

### 5.1 Inference-time reranking with reciprocal nearest neighbors

We first evaluate the effectiveness of reciprocal nearest neighbors at improving the similarity metric between queries and documents.

Across all query sets in two important evaluation settings, MS MARCO (Table 1) and TripClick (Table 2), we observe that using a similarity based on reciprocal nearest neighbors can consistently improve ranking effectiveness for all tested models. The magnitude of improvement is generally small, but becomes substantial when measured on the TREC DL datasets (approx. +0.010 nDCG@10), where a greater annotation depth and multi-level relevance labels potentially allow to better differentiate between methods.

We furthermore observe that ranking effectiveness initially improves when increasing the size of the ranking context (i.e., the number of candidates considered for reranking), which is expected, because the probability to include a remote ground-truth document in the context increases. However, as this size further increases, ranking effectiveness saturates, often peaking at a context size of a few tens of candidates (Figures 3, 4, 6, 9). We hypothesize that this happens because, as we keep adding negatives in the context, the chance that they disrupt the reciprocal neighborhood relationship between query and positive document(s) increases (see Figure 1).

We therefore conclude that we may use a relatively small number $N$ of context candidates for computing reciprocal nearest neighbor similarities, which is convenient because computational com-

| Model | TREC DL 2019 | | | | TREC DL 2020 | | | |
|---|---|---|---|---|---|---|---|---|
| | MRR | nDCG | MAP | Recall | MRR | nDCG | MAP | Recall |
| TAS-B | 0.875 | 0.659 | 0.222 | 0.259 | 0.832 | 0.620 | 0.302 | 0.363 |
| CODER(TAS-B) | 0.857 | 0.668 | 0.224 | 0.270 | 0.844 | 0.623 | 0.306 | 0.365 |
| CODER(TAS-B) + uniform sm. | 0.857 | 0.669 | 0.223 | 0.273 | 0.835 | 0.619 | 0.304 | 0.360 |
| CODER(TAS-B) + geom. smooth labels | 0.848 | 0.665 | 0.220 | 0.271 | 0.842 | 0.626 | 0.310 | 0.370 |
| CODER(TAS-B) + rNN smooth labels | 0.857 | 0.671 | 0.226 | 0.276 | **0.862** | 0.632 | 0.315 | 0.369 |
| **CODER(TAS-B) + mixed rNN/geom. smooth lab.** | **0.889** | **0.675** | **0.227** | **0.277** | 0.842 | **0.637** | **0.318** | **0.376** |
| CoCondenser | 0.879 | 0.656 | 0.226 | 0.269 | 0.833 | 0.618 | 0.301 | 0.366 |
| CODER(CoCondenser) | **0.895** | 0.655 | 0.228 | 0.269 | 0.844 | 0.639 | 0.314 | **0.384** |
| **CODER(CoCondenser) + mixed rNN/geom. smooth lab.** | 0.884 | **0.661** | **0.232** | **0.278** | **0.856** | **0.646** | **0.316** | 0.383 |

Table 3: Evaluation of label smoothing applied to training CODER(TAS-B) on MS MARCO. Metrics cut-off @10. **Bold**: best performance in each model class. Blue: our contributions.

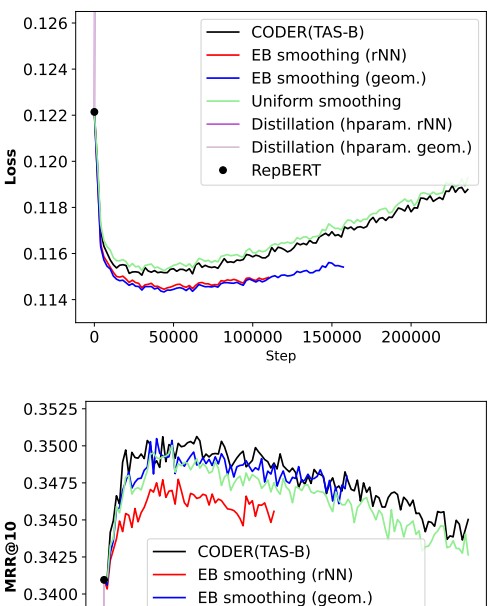

Figure 2: Evolution of performance of TAS-B (left-most point, step 0) on MS MARCO dev validation set, as the model is being fine-tuned through CODER. The red curve corresponds to using evidence-based (EB) label smoothing computed with rNN-based similarity, whereas for the blue curve the smooth label distribution is computed using pure geometric similarity. EB label smoothing significantly reduces validation loss (computed with the original labels, top), indicating that the ground truth passages are receiving higher probability in the estimated relevance distribution, but the retrieval metric (bottom) fails to register an improvement due to annotation sparsity (compare with Fig. 10, Appendix). Distillation leads to precipitous degradation of performance.

plexity scales with $O(N^2)$. In MS MARCO, a context of 60 candidates corresponds to peak effectiveness for CODER(TAS-B) and introduces a CPU processing delay of only about 5 milliseconds per query (Figure 7). We expect the optimal context size to depend on the average rank that ground-truth documents tend to receive, and for models of similar ranking effectiveness, this would primarily be determined by the characteristics of the dataset.

Indeed, we find that the hyperparameters in computing rNN-based similarity (e.g. $k$, $\lambda$, $\tau$, $f_w$), as well as the context size $N$, predominantly depend on the dataset, and to a much lesser extent on the dense retriever: hyperparameters optimized for CODER(TAS-B) worked very well for TAS-B, CoCondenser and CODER(CoCondenser) on MS MARCO, but very poorly when transferred to TripClick.

A more detailed description and discussion of reranking experiments is provided in Appendix A.4.

## 5.2 Evidence-based label smoothing

In order to achieve the best possible results using evidence-based label smoothing, one should ideally optimize the hyperparameters related to rNN-based similarity for the specific task of training a retrieval model with recomputed soft labels. However, to avoid repeatedly computing soft labels for the training set, we simply chose an rNN configuration that was optimized for reranking a large pool of candidates ($N = 1000$) in the MS MARCO collection – i.e., the same one used in the previous section. Although this configuration may not be optimal for our specific task (e.g., small changes in score values might be sufficient for reranking candidates but ineffective as soft training labels), we expect that it can still provide a reliable lower bound of optimal performance.

Figure 2 shows how the ranking performance of the TAS-B base model (left-most point, step 0) on the validation set evolves throughout fine-tuning through CODER. The red curve corresponds to additionally using evidence-based label smooth-

ing computed with reciprocal NN-based similarity (rNN-related hyperparameters are the same as in Section 5.1), whereas for the blue curve the smooth label distribution is computed using pure geometric similarity. We observe the following seemingly paradoxical phenomenon: compared to plain CODER training, label smoothing significantly reduces the *validation* loss (computed with the original labels, top panel), indicating that the ground truth passages are now receiving proportionally higher scores in the estimated relevance distribution, but the retrieval metric (bottom panel) does not register an improvement.

In fact, this phenomenon may be fully explained through the presence of false negatives: through the smooth target label distribution, the model learns to assign high relevance scores to a larger number of documents (diffuse distribution). Therefore, it likely places a proportionally higher relevance distribution weight to the ground truth document compared to plain CODER, essentially improving the relevance estimate for the ground truth, but at the same time it distributes relevance weight to a higher number of candidates, such that the ground truth ends up being ranked slightly lower (see Figure 11).

The crucial question therefore is, whether the candidates now receiving a higher relevance score are actually relevant. Since the MS MARCO dev dataset almost always contains only a single positive-labeled passage per query, it is fundamentally ill-suited to measure ranking effectiveness improvements by a training scheme that primarily promotes a diffuse relevance distribution over several candidates.

For this reason, we must rely on datasets containing more judgements per query, such as the TREC DL 2019, 2020 datasets: Table 3 shows that evidence-based label smoothing using a similarity based on reciprocal nearest neighbors can significantly improve the performance of each dense retriever even beyond the benefit of the plain CODER fine-tuning framework. Furthermore, using an rNN-based Jaccard similarity as a metric for computing the soft labels yields significantly better performance than using geometric similarity, and the best results are achieved when using a linear combination of the two metrics.

As TripClick also contains several (pseudo-relevance) labels per query, we additionally evaluate the MS MARCO-trained models zero-shot

| Model | DCTR Head | | | RAW Head | | |
|---|---|---|---|---|---|---|
| | MRR | nDCG | Recall | MRR | nDCG | Recall |
| RepBERT | 0.526 | 0.255 | 0.242 | 0.574 | 0.344 | 0.199 |
| CODER(RB) | 0.610 | 0.300 | 0.276 | 0.656 | 0.401 | 0.228 |
| CODER(RB) hparam. | 0.608 | 0.300 | 0.277 | 0.649 | 0.401 | 0.229 |
| **CODER(RB) + EB smooth.** | **0.611** | **0.305** | **0.280** | **0.661** | **0.411** | **0.234** |

Table 4: Evaluation of evidence-based label smoothing (mixed rNN - geom. similarity) on TripClick HEAD Test. Models were trained on TripClick HEAD ∪ TORSO Train and validated on HEAD Val. Metrics cut-off @10. "hparam": model trained with same hyperparameters as the one with label smoothing. Blue: our contributions.

(i.e., without any training) on TripClick Test and Val (Figures 7, 8, Appendix). We again observe that evidence-based label smoothing with an rNN-based metric improves performance compared to plain CODER; however, we note that in this zero-shot setting, the best performing models were not in general the same as the best performing models on TREC DL. The best ranking performance was achieved by CODER(TAS-B) using soft labels from pure rNN-based Jaccard similarity.

We thus find that in sparsely annotated datasets like MS MARCO, validation loss might be a better predictor of model generalization than IR metrics such as MRR, and that evaluation on datasets with higher annotation depth (such as TREC DL or TripClick), potentially even in a zero-shot setting, might better reflect the ranking effectiveness of models.

A critical difference of evidence-based label smoothing from distillation is that soft document labels are computed based on their similarity to the ground truth instead of the query. To demonstrate the importance of this change of perspective, we show how CODER fine-tuning performs when using soft labels coming from geometric similarity with respect to the query, as in distillation (Figure 2, purple curves): even when applying the same transformations to the scores as in the case of evidence-based label smoothing, the model's performance rapidly degrades instead of improving. This is expected, because distillation only works when a superior model is available; training cannot be bootstrapped from the scores of the model itself.

We also observe that, unlike evidence-based label smoothing, uniform label smoothing fails to noticeably improve performance compared to plain CODER fine-tuning (Figure 2, Table 3), even when we ensure that the exact same probability weight as in the case of evidence-based smoothing is distributed from the ground-truth positive(s) among

the rest of the candidates.

Finally, we examine how EB label smoothing performs when training in an important alternative setting, TripClick: a dataset with significantly more relevance labels per query, that come from pseudo-relevance feedback without human judgements. Unlike above, here we investigate the joint optimization of rNN-related parameters together with training-specific parameters (e.g., learning rate and linear warm-up steps), instead of using the same rNN-related hyperparameters for label smoothing as for reranking. To allow this, we train on the union of the HEAD and TORSO training subsets (avg. 42 and 9 annotations per query, respectively), and omit the TAIL subset, which consists of a large number of rare queries (each with only 3 annotations on average). We use HEAD Val as a validation set, and evaluate on HEAD Test.

Table 4 and Figure 10 show that training with mixed geometric/rNN-based smooth labels significantly improves performance also in this dataset setting compared to plain CODER training (+0.010 nDCG@10). To ensure that any improvement cannot be attributed to better hyperparameters found during the joint optimization described above, we also apply the same hyperparameters to plain CODER training (denoted "hyperparam." in the table). We observe similar improvements on TORSO Test and TORSO Val (Appendix Table 9).

## 6 Conclusion

We propose evidence-based label smoothing to address sparse annotation in dense retrieval datasets. To mitigate penalizing the model in case of false negatives during training, we compute the target relevance distribution by assigning non-zero relevance probabilities to candidates most similar to the ground truth. To estimate similarity we leverage reciprocal nearest neighbors, which allows considering local connectivity in the shared representation space, and can independently be used for reranking. Extensive experiments on two large-scale retrieval datasets and three dense retrieval models demonstrate that our method can effectively improve ranking, while being computationally efficient and foregoing the use of resource-heavy cross-encoders. Finally, we find that evaluating on sparsely annotated datasets like MS MARCO dev may systematically underestimate models with less sharp (i.e. more diffuse) relevance score distributions.

## Acknowledgements

G. Zerveas would like to thank the Onassis Foundation for supporting this research. The contribution of N. Rekabsaz is supported by the State of Upper Austria and the Federal Ministry of Education, Science, and Research, through grant LIT-2021-YOU-215.

## Limitations

We believe that in principle, the methods we propose are applicable to any dual-encoder dense retriever: computing the similarity metric based on reciprocal nearest neighbors only requires access to the encoder extracting the document and query embeddings.

However, we note that the reason we were able to compute the soft labels for evidence-based label smoothing completely offline was that we utilized CODER as a fine-tuning framework: CODER only fine-tunes the query encoder, using fixed document representations. Using evidence-based label smoothing in a training method with learnable document embeddings means that the rNN-based similarity has to be computed dynamically at each training step (or periodically every few training steps), because their mutual distances/similarities will change during training, albeit slowly. Similarly, every time candidates/negatives are retrieved dynamically (periodically, as in Xiong et al., 2020, or at each step, as in Zhan et al., 2021) the rNN-based similarity has to be recomputed among this new set. Nevertheless, as we discuss in the paper, we only need to use a context of tens or at most a couple of hundred candidates in order to compute the rNN-based similarity most effectively. Even in these cases, this would therefore introduce at most up to a hundred milliseconds of training delay per batch, while inference would remain unaffected.

## Ethics Statement

By being computationally efficient and foregoing the use of resource-heavy cross-encoders in its pipeline, our method allows top-performing dense retrieval models to be fine-tuned on MS MARCO within 7 hours on a single GPU. We therefore believe that it is well-aligned with the goal of training models in an environmentally sustainable way, the importance of which has been recently acknowledged by the scientific community Information Retrieval and more broadly (Scells et al., 2022).

On the other hand, the transformer-based Information Retrieval models examined in our study may intrinsically exhibit societal biases and stereotypes. As prior research has discussed (Gezici et al., 2021; Rekabsaz et al., 2021a; Rekabsaz and Schedl, 2020; Bigdeli et al., 2022; Krieg et al., 2022; Bigdeli et al., 2021; Fabris et al., 2020), these biases stem from the latent biases acquired by transformer-based language models throughout their pre-training, as well as the fine-tuning process on IR collections. Consequently, the practical use of these models might result in prejudiced treatment towards various social groups (e.g., as manifested in their representation or ranking in retrieval result lists). We therefore firmly encourage a mindful and accountable application of these models.

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

## A Appendix

### A.1 Jaccard similarity based on Reciprocal Nearest Neighbors

Let $\mathcal{C}$ be a collection of documents, including the query used for search, and $\mathrm{NN}(q, k)$ denote the set of $k$-nearest neighbors of a probe $q \in \mathcal{C}$ – besides the query, $q$ here can also be a document or any other element that can be embedded in the common representation space. If $d(q, c_i) \equiv d_g(\mathbf{x}_q, \mathbf{x}_{c_i})$, $c_i \in \mathcal{C}$ is a metric (distance) in the vector space within which the embeddings of the query $\mathbf{x}_q$ and documents $\mathbf{x}_i$ reside, we can formally write:

$$\mathrm{NN}(q, k) = \{c_i \mid d_g(\mathbf{x}_{c_i}, \mathbf{x}_q) \le d_g(\mathbf{x}_{c_k}, \mathbf{x}_q), \\ \forall i \in \mathbb{N} : 1 \le i \le |\mathcal{C}|\}, \quad (3)$$

where $|\cdot|$ denotes the cardinality of a set, and document $c_k$ is the $k$-nearest neighbor of the query based on $d$, i.e., the $k$-th element in the list of all documents in $\mathcal{C}$ sorted by distance $d$ from the query in ascending order[5]. Naturally, $|\mathrm{NN}(q, k)| = k$.

The set of $k$-reciprocal nearest neighbors can then be defined as:

$$\mathcal{R}(q, k) = \{c_i \mid c_i \in \mathrm{NN}(q, k) \wedge q \in \mathrm{NN}(c_i, k)\}, \quad (4)$$

i.e., to be considered a $k$-reciprocal neighbor, a document must be included in the $k$-nearest neighbors of the query, but at the same time the query must also be included in the $k$-nearest neighbors of the same document. This stricter condition results in a stronger similarity relationship than simple nearest neighbors, and $|\mathcal{R}(q, k)| \le k$.

Since using the above definition as-is can be overly restrictive, prior work has proposed applying it iteratively in order to construct an extended set of highly related documents to the query that would have otherwise been excluded. Thus, Zhong et al. (2017) define the extended set:

$$\mathcal{R}^*(q, k) := \mathcal{R}(q, k) \cup \mathcal{R}(c_i, \tau k), \\ \text{s.t. } \left|\mathcal{R}(q, k) \cap \mathcal{R}(c_i, \tau k)\right| \ge \frac{2}{3}\left|\mathcal{R}(c_i, \tau k)\right|, \quad (5) \\ \forall c_i \in \mathcal{R}(q, k).$$

---

[5] When some measure of similarity $s$ is used instead of a distance $d$, the relationship equivalently becomes: $s(\mathbf{x}_{c_i}, \mathbf{x}_q) \ge s(\mathbf{x}_{c_k}, \mathbf{x}_q)$, and the $k$-nearest neighbors are the first $k$ documents sorted by $s$ in descending order.

Effectively, we examine the set of $\tau k$-nearest reciprocal neighbors of each reciprocal neighbor of $q$ (where $\tau \in [0, 1]$ is a real parameter), and provided that it already has a substantial overlap with the original set of reciprocal neighbors of $q$, we add it to the extended set. The underlying assumption is that if a document is closely related to a set of documents that are closely related to the query, then it is most likely itself related to the query, even if there is no direct connection in terms of geometric proximity. Thus, one can improve recall at the possible expense of precision.

Although using this new set of neighbors as the new set of candidates and sorting them by their distance $d$ can form the basis of a retrieval method, Zhong et al. (2017) additionally proceed to define a new distance that takes into account this set, which is used alongside $d$. Specifically, they use the Jaccard distance between the (extended) reciprocal neighbor sets of a query $q$ and documents $c_i$:

$$d_J(q, c_i) = 1 - \frac{\left|\mathcal{R}^*(q, k) \cap \mathcal{R}^*(c_i, k)\right|}{\left|\mathcal{R}^*(q, k) \cup \mathcal{R}^*(c_i, k)\right|}. \quad (6)$$

This distance quantifies similarity between two elements (here, $q$ and $c_i$) as a measure of overlap between sets of neighbors robustly related to each of them.

To reduce the computational complexity of computing the Jaccard distance, which relies on the time-consuming, CPU-bound operations of finding the intersection and union of sets, one may instead carry out the computation with algebraic operations, by defining for each element $q \in \mathcal{C}$ sparse vectors of dimensionality $|\mathcal{C}|$, where non-zero dimensions denote graph connectivity to other documents. Instead of using binary vectors, one may assign to each neighbor $c_i$ a weight that depends on its geometric distance to the probe $q$. Thus, following Zhong et al. (2017), we define the elements of reciprocal connectivity vectors $\mathbf{v}'_q \in |\mathcal{C}|$ as follows:

$$v'_{q,c_i} = \begin{cases} f_w\left(d(q, c_i)\right) & \text{if } c_i \in \mathcal{R}^*(q, k) \\ 0 & \text{otherwise} \end{cases} \quad (7)$$

While Zhong et al. (2017) exclusively use $f_w(x) = \exp(-x)$, one one can use any monotonically decreasing function, and we found that $f_w(x) = -x$ in fact performs better in our experiments.

Finally, instead of directly using the sparse vectors above, which would yield a discretized similarity metric, we perform a local expansion, mixing each one of them (including the query) with its $k_{\text{exp}}$ neighboring vectors (again including the query, if among the neighbors):

$$\mathbf{v}_{c_i} = \frac{1}{k_{\text{exp}}} \sum_{j=1}^{k_{\text{exp}}} \mathbf{v}'_{c_j} \, , \; \forall c_j \in \text{NN}(c_i, k_{\text{exp}}) \, . \quad (8)$$

It is possible to use the element-wise $\min$ and $\max$ operators on the expanded sparse vectors from Eq. (8)) to compute the number of candidates in the intersection and union sets of Eq. (6) respectively as:

$$\left| \mathcal{R}^*(q,k) \cap \mathcal{R}^*(c_i,k) \right| = \sum \min(\mathbf{v}_q, \mathbf{v}_{c_i}) \quad (9)$$

$$\left| \mathcal{R}^*(q,k) \cup \mathcal{R}^*(c_i,k) \right| = \sum \max(\mathbf{v}_q, \mathbf{v}_{c_i}), \quad (10)$$

and thus the Jaccard distance in Eq. (6) can be written as:

$$d_J(q, c_i) = 1 - \frac{\sum_{j=1}^{|\mathcal{C}|} \min(v_{q,c_j}, v_{c_i,c_j})}{\sum_{j=1}^{|\mathcal{C}|} \max(v_{q,c_j}, v_{c_i,c_j})}. \quad (11)$$

Finally, we note that instead of the pure Jaccard distance $d_J$, we use as the final distance $d^*$ a linear mixing between the geometric distance $d$ and $d_J$ with a hyperparameter $\lambda \in [0, 1]$:

$$d^*(q, c_i) = \lambda d(q, c_i) + (1 - \lambda) d_J(q, c_i), \quad (12)$$

which we found to perform better both for reranking (as in Zhong et al., 2017), as well as for label smoothing.

## A.2 Data

### A.2.1 MS MARCO and TREC Deep Learning

Following the standard practice in related contemporary literature, we use the MS MARCO dataset (Bajaj et al., 2018), which has been sourced from open-domain logs of the Bing search engine, for training and evaluating our models. The MS MARCO passage collection contains about 8.8 million passages and the training set contains about 503k queries labeled with one or (rarely) more relevant passages (1.06 passages per query, on average), on a single level of relevance.

For validation of the trained models we use a subset of 10k samples from "MS MARCO dev", which is a set containing about 56k labeled queries, and refer to it as "MS MARCO dev 10k". As a test set we use a different, officially designated subset of "MS MARCO dev", originally called "MS MARCO dev.small", which contains 6980 queries. Often, in literature and leaderboards it is misleadingly referred to as "MS MARCO dev".

Because of the very limited annotation depth (sparsity) in the above evaluation sets, we also evaluate on the TREC Deep Learning track 2019 and 2020 test sets, each containing 43 and 54 queries respectively, labeled to an average "depth" of more than 210 document judgements per query, and using 4 levels of relevance: "Not Relevant" (0), "Related" (1), "Highly Relevant" (2) and "Perfect" (3). According to the official (strict) interpretation of relevance labels[6], a level of 1 should not be considered relevant and thus be treated just like a level of 0, while the lenient interpretation considers passages of level 1 relevant when calculating metrics.

### A.2.2 TripClick

TripClick is a recently introduced health IR dataset (Rekabsaz et al., 2021b) based on click logs that refer to about 1.5M MEDLINE articles. The approx. 700k unique queries in its training set are split into 3 subsets, HEAD, TORSO and TAIL, based on their frequency of occurrence: queries in TAIL are asked only once or a couple of times, while queries in HEAD have been asked tens or hundreds of times. As a result, each query in HEAD, TORSO and TAIL on average ends up with 41.9, 9.1 and 2.8 pseudo-relevance labels, using a click-through model (RAW) where every clicked document is considered relevant. The dataset also includes alternative relevance labels using the Document Click-Through Rate (DCTR), on 4 distinct levels (the latter follow the same definitions as the TREC Deep Learning evaluation sets). We note that, although the number of labels per query is much higher than MS MARCO, unlike the latter, these labels have not been verified by human judges.

For validation and evaluation of our models we use the officially designated validation and test set, respectively (3.5k queries each).

---

[6]https://trec.nist.gov/data/deep2019.html

### A.3 Evaluation

All training and evaluation experiments are produced with the same seed for pseudo-random number generators. We use mean reciprocal rank (MRR), normalized discounted cumulative gain (nDCG), mean average precision (MAP) and recall to evaluate the models on TREC DL tracks, MS MARCO and TripClick, in line with past work (e.g. (Xiong et al., 2020; Zhan et al., 2021; Hofstätter et al., 2021; Rekabsaz et al., 2021b)). While relevance judgements are well-defined in MS MARCO and TripClick, for the TREC DL tracks there exist strict and lenient interpretations of the relevance scores of judged passages (see Section A.2). In this work, we use the official, strict interpretation. We calculate the metrics using the official TREC evaluation software.[7]

---

[7]`trec.nist.gov/trec_eval/index.html`

## A.4 Inference-time reranking with reciprocal nearest neighbors

Prior work on rNN reranking considered the entire gallery of images (collection $\mathcal{C}$) as a reranking context for each probe, i.e. $N = |\mathcal{C}|$. With $|\mathcal{C}|$ in the order of tens of millions, this is intractable for the task of web retrieval using transformer LMs, and a smaller context size must be used instead. To investigate the importance of the context size, we therefore initially fix the number of in-context candidates per query to a large number within reasonable computational constraints ($N = 1000$) and optimize the hyperparameters of reciprocal nearest neighbors (e.g. $k$, $k_{\exp}$, $\lambda$, $\tau$, $f_w$) on the MS MARCO dev.small subset.

We first rerank candidates initially ranked by a CODER-optimized TAS-B retriever, denoted as "CODER(TAS-B)". To determine an appropriate size of reranking context, we first sort candidates by their original relevance score (geometric similarity) and then recompute query similarity scores with a growing number of in-context candidates (selected in the order of decreasing geometric similarity from the query), while measuring changes in ranking effectiveness.

Figure 3 shows that rNN-based reranking slightly improves effectiveness compared to ranking purely based on geometric similarity, with the peak improvement registered around a context size of 60 candidates. This behavior is consistent when evaluating rNN-based raranking using the same hyperparameters on different query sets: MS MARCO dev (Fig. 4), which is an order of magnitude larger, and TREC DL 2020 (Fig. 5) and TREC DL 2019 (Fig. 6), where the improvement is larger (possibly because it can be measured more reliably due to the greater annotation depth). In all cases performance clearly saturates as the number of candidates grows (somewhat slower for TREC DL 2019). The same behavior as described above is observed when reranking the original TAS-B model's results using the same hyperparameters chosen for the CODER-trained version, with the performance benefit being approximately twice as large (Fig. 8).

While it is expected that progressively increasing the context size will increase performance, as there is a greater chance to include the ground-truth passage(s) which may have been initially ranked lower (i.e. embedded farther from the query), the peak and subsequent degradation or saturation is a novel finding. We hypothesize that it happens because the more negative candidates are added in the context, the higher the chance that they disrupt the reciprocal neighborhood relationship between query and positive document(s) (see Figure 1).

We can therefore conclude that we may use a relatively small number $N$ of context candidates for computing reciprocal nearest neighbors similarities, which is convenient because computational complexity scales with $O(N^2)$. For a context of 60 candidates, a CPU processing delay of only about 5 milliseconds per query is introduced (Figure 7). These results additionally indicate that the context size should best be treated as a rNN hyperparameter to be jointly optimized with the rest, which is reasonable, as it is expected to depend on the average rank that ground-truth documents tend to receive.

After optimizing rNN-related hyperparameters (including the context size) on MS MARCO dev.small for CODER(TAS-B), we evaluate rNN reranking on the other evaluation sets (including its $\times 8$ larger superset MS MARCO dev) and present the results in Table 1. We observe that a similarity based on reciprocal nearest neighbors can indeed improve ranking effectiveness compared to using purely geometric similarity. The improvement is more pronounced on the TREC DL datasets (+0.011 nDCG@10), where a greater annotation depth and multi-level relevance labels potentially allow to better differentiate between methods.

Additionally, we find that rankings from TAS-B – whose embeddings are relatively similar to CODER(TAS-B) – also improve, despite the fact that hyperparameters were chosen based on the CODER(TAS-B) model (also see Figure 8).

The strongest dense retrieval models we evaluate, CoCondenser and CODER(CoCondenser), also show improved performance, again measured primarily on TREC DL: the former improves by +0.009 nDCG@10 on TREC DL 2020 and the latter by 0.009 nDCG@10 on TREC DL 2019. Notably, reranking effectiveness when using the exact same hyperparameters as for CODER(TAS-B) and TAS-B is only very slightly worse.

By contrast, when transferring hyperparameters selected for MS MARCO to reranking candidates on the TripClick dataset, we find that performance deteriorates with respect to geometric similarity. Therefore, we can conclude that rNN hyperparameters predominantly depend on the dataset, and to

| Hyperparameter | TAS-B | CODER(TAS-B) | CoCondenser | CODER(CoCondenser) |
|---|---|---|---|---|
| $N_r$: context size | 60 | 60 | 53 | 63 |
| $k$: num. NN | 21 | 21 | 21 | 19 |
| $k_{exp}$: num. NN for expansion | 3 | 3 | 5 | 8 |
| $\tau$: trust factor | 0 | 0 | 0.128 | 0.5 |
| $\lambda$: linear comb. coeff. | 0.451 | 0.451 | 0.469 | 0.473 |

Table 5: Hyperparameters for reranking with Reciprocal Nearest Neighbors, MS MARCO.

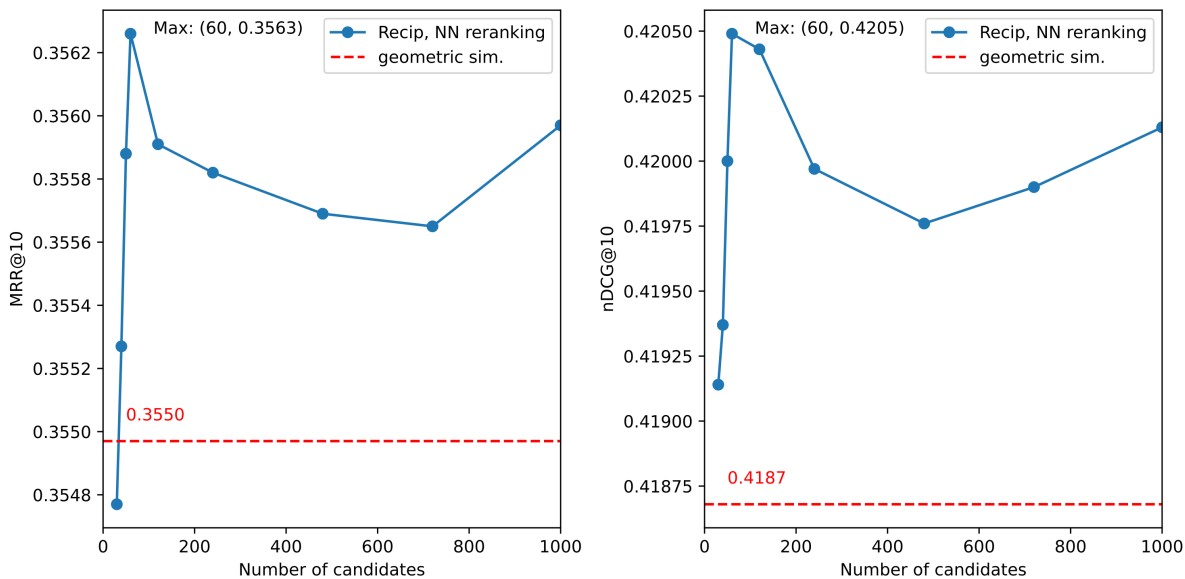

Figure 3: Performance of reciprocal nearest neighbors-based reranking of CODER(TAS-B) results on MS MARCO dev.small, as the number of candidates in the ranking context grows. Hyperparameters are optimized for a context of 1000 candidates. Performance is slightly improved compared to ranking exclusively based on geometric similarity and peaks at 60 in-context candidates.

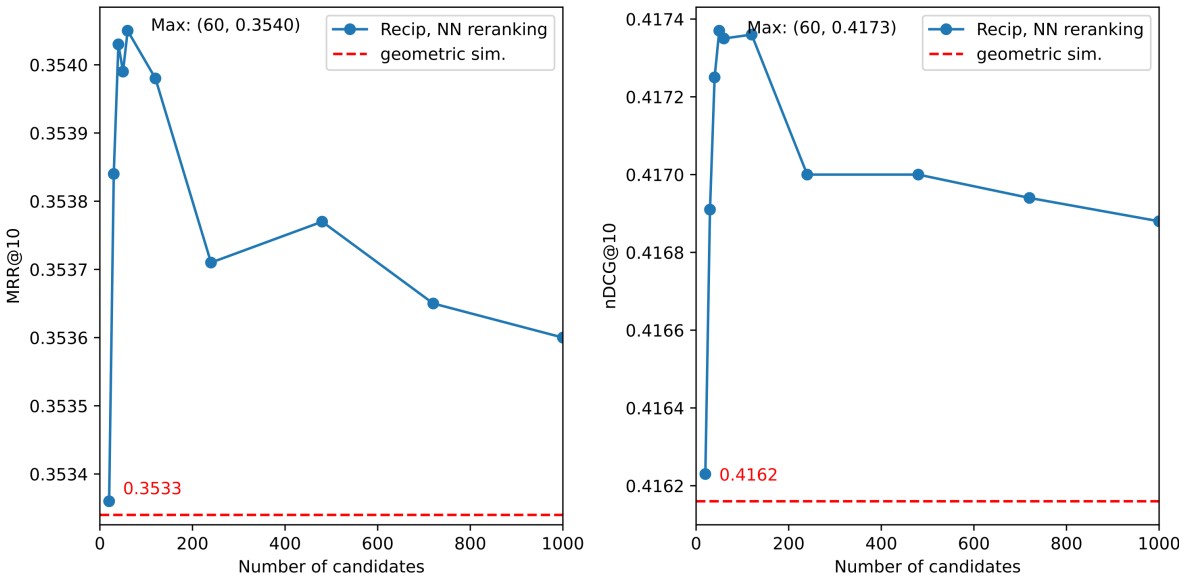

Figure 4: Performance of reciprocal nearest neighbors-based reranking of CODER(TAS-B) results on MS MARCO dev, as the number of candidates in the ranking context grows. Hyperparameters are the same as in Fig. 3. Performance is slightly improved compared to ranking exclusively based on geometric similarity and peaks at 60 in-context candidates.

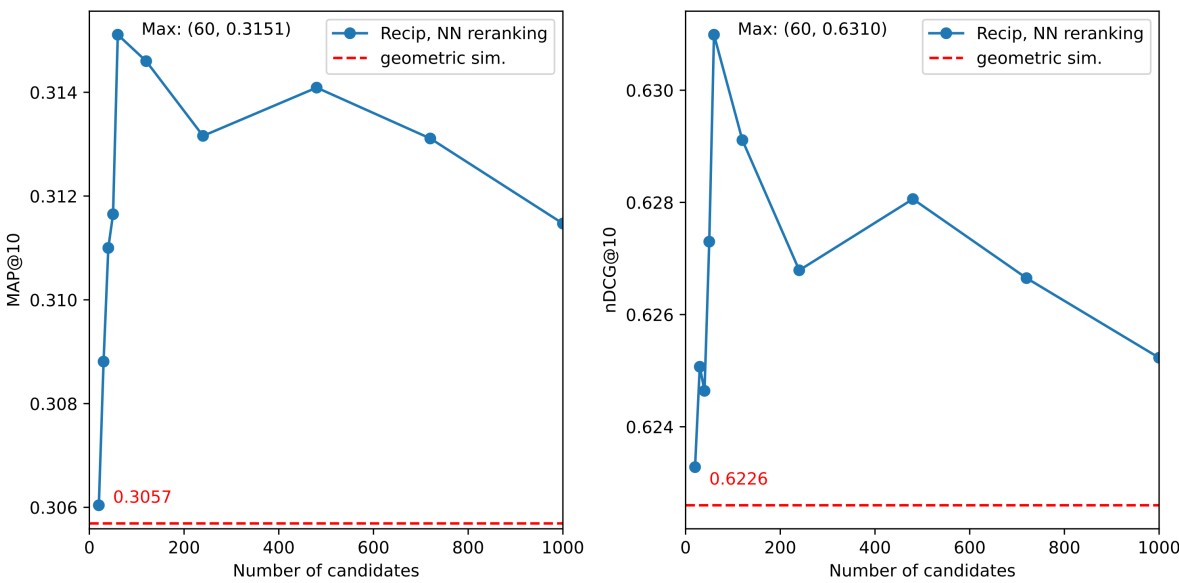

Figure 5: Performance of reciprocal nearest neighbors-based reranking of CODER(TAS-B) results on TREC DL 2020, as the number of candidates in the ranking context grows. Hyperparameters are the same as in Fig. 3. Performance is improved compared to ranking exclusively based on geometric similarity and peaks at 60 in-context candidates.

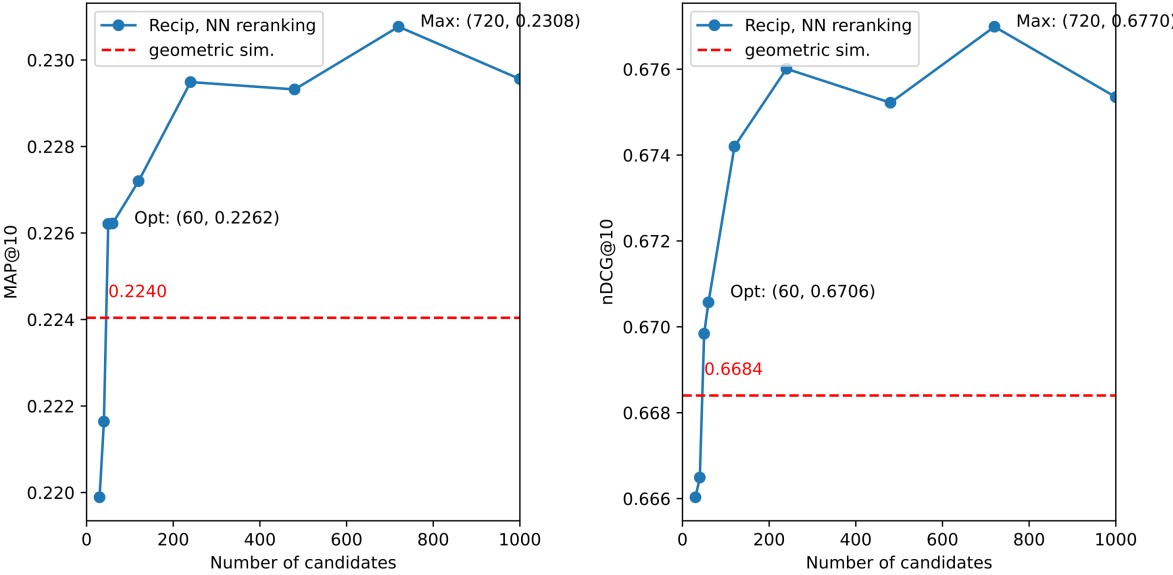

Figure 6: Performance of reciprocal nearest neighbors-based reranking of CODER(TAS-B) results on TREC DL 2019, as the number of candidates in the ranking context grows. Hyperparameters are the same as in Fig. 3. Performance is improved compared to ranking exclusively based on geometric similarity but does not clearly saturate.

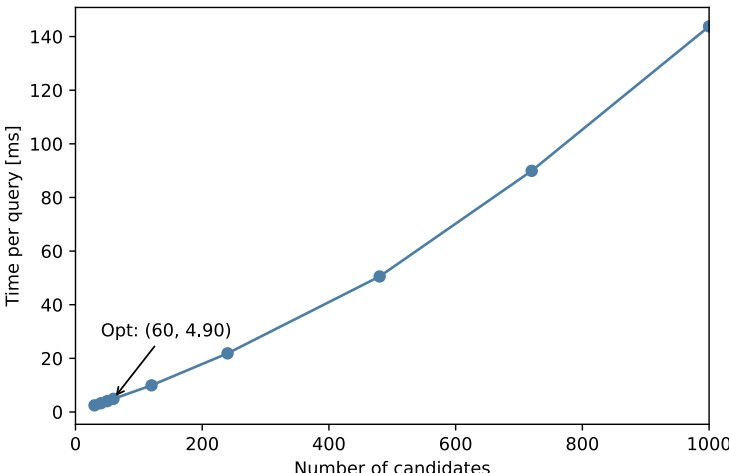

Figure 7: Time delay per query (in milliseconds) when reranking using reciprocal nearest neighbors-based similarity, as the number of candidates in the ranking context grows. Hyperparameters are the same as in Fig. 3. Processing time scales according to $O(N^2)$. Processor (1 CPU, 1 core): AMD EPYC 7532 32-Core Processor, 2400 MHz.

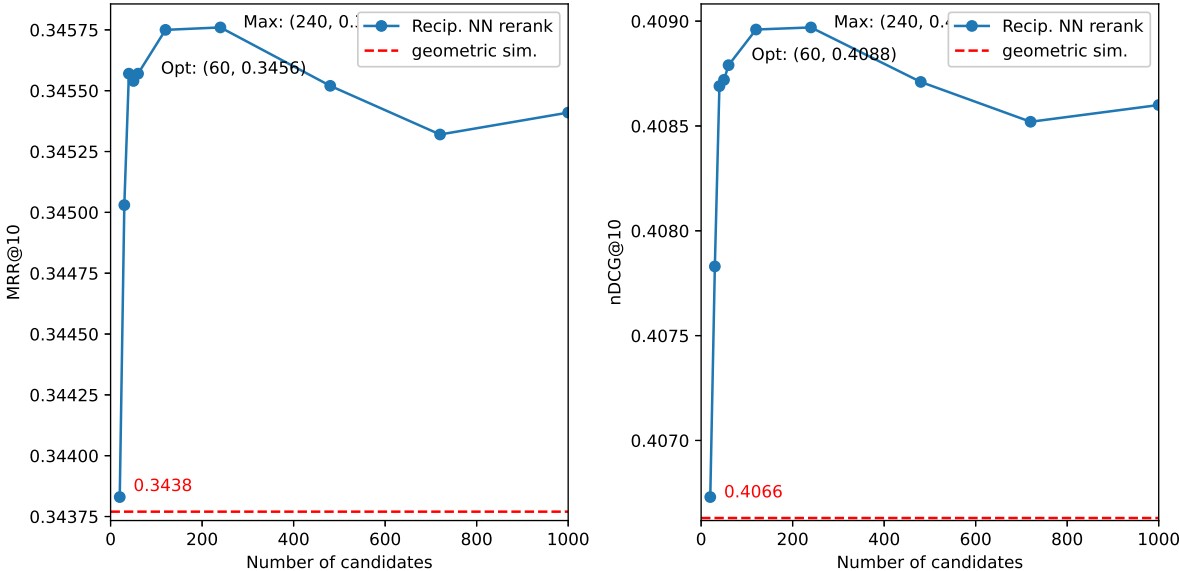

Figure 8: Performance of reciprocal nearest neighbors-based reranking of TAS-B results on MS MARCO dev, as the number of candidates in the ranking context grows. Hyperparameters are the same as in Fig. 3. Performance is improved compared to ranking exclusively based on geometric similarity and peaks at approx. 60 in-context candidates.

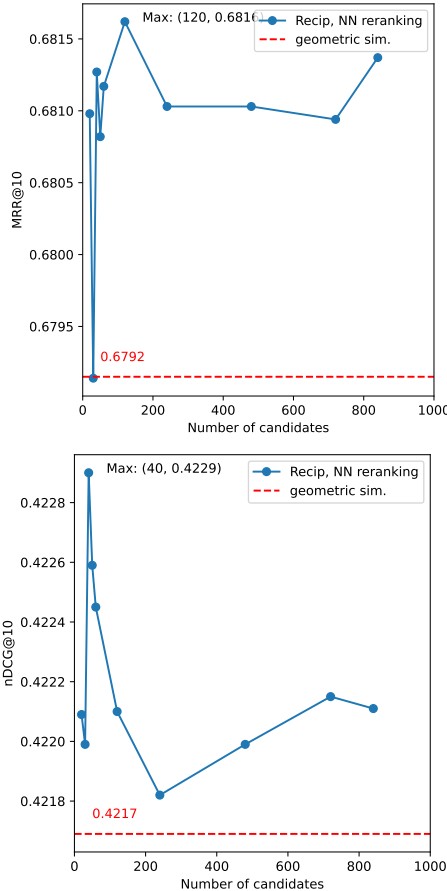

Figure 9: Performance of reciprocal nearest neighbors-based reranking of CODER(RepBERT) results on TripClick `HEAD Test`, as the number of candidates in the ranking context grows.

a much lesser extent on the dense retriever.

After optimizing hyperparameters on TripClick `HEAD Val`, we evaluate on `HEAD Test`, using both RAW (binary) as well as DCTR (multi-level) relevance labels; we present the results in Table 2. Also for this dataset, which differs substantially in characteristics from MS MARCO, we again observe that using reciprocal nearest neighbors to compute the similarity metric can slightly improve ranking effectiveness for all examined retrieval methods. We also observe the same saturation behavior with respect to the ranking context size, i.e. the number of candidates considered when reranking (Fig. 9).

## A.5 Evidence-based label smoothing

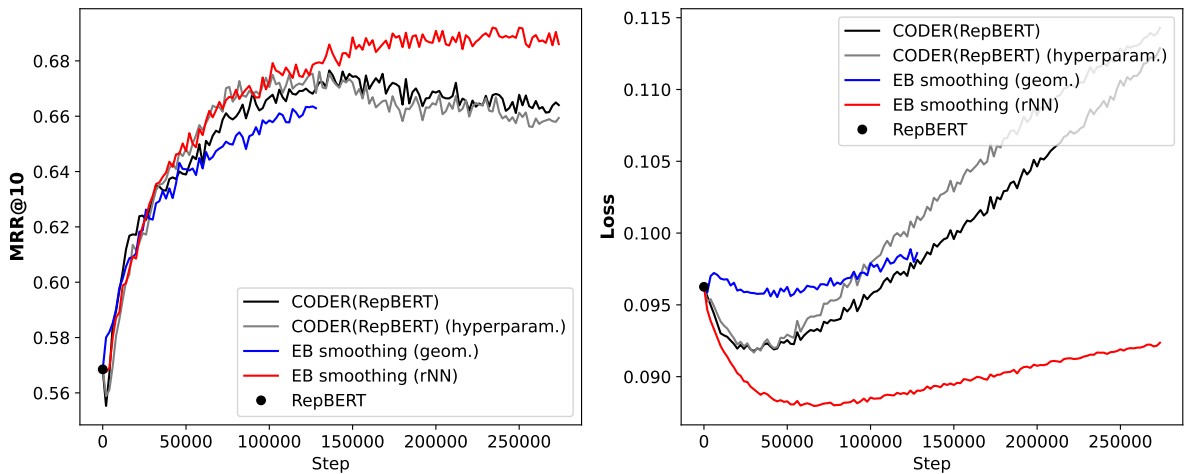

Figure 10: Evolution of performance of RepBERT (left-most point, step 0) on the TripClick `HEAD Val` validation set, as the model is being fine-tuned through CODER on TripClick `HEAD⊔TORSO Train`. The red curve corresponds to additionally using evidence-based label smoothing computed with reciprocal NN-based similarity, whereas for the blue curve the smooth label distribution is computed using pure geometric similarity. Only evidence-based smoothing with rNN similarity substantially improves performance compared to plain CODER(RepBERT), despite "CODER(RepBERT) (hyperparam.)" and "EB smoothing" with geometric similarity using the same training hyperparameters.

| EB label smoothing hyperparam. | CODER(TAS-B) | CODER(CoCondenser) |
|---|---|---|
| $b$: boost factor | 1.222 | 1.525 |
| $n_{max}$: softmax cut-off | 4 | 32 |
| $f_n$: normalization func. | max-min | std-based |
| learning rate: peak value | 1.73e-06 | 1.37e-06 |
| learning rate: linear warm-up steps | 9000 | 12000 |

Table 6: Hyperparameters for training with evidence-based label smoothing, MS MARCO. The hyperparameters related to computing rNN-based similarity are the same as in Table 5.

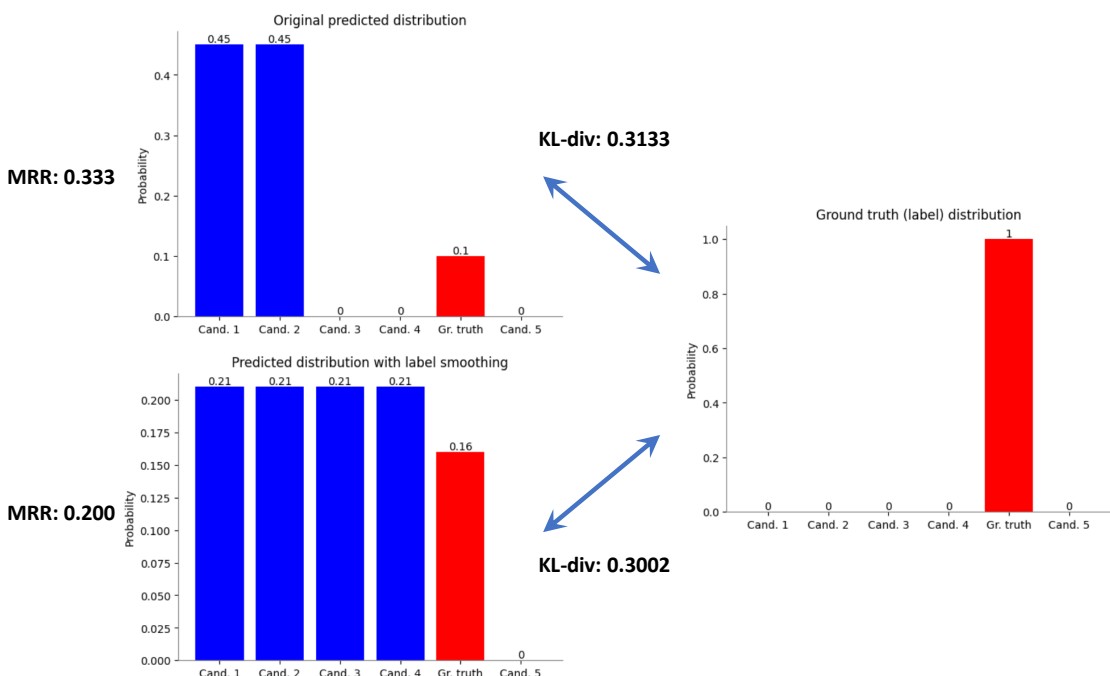

Figure 11: Because many more documents receive higher than zero relevance in the target distribution after label smoothing, by design it promotes a diffuse predicted distribution (bottom). Thus, although the predicted relevance of the ground-truth positive document is now significantly higher compared to when not using label smoothing (top), indicating a model improvement, the document ends up ranking lower because of the dispersed relevance estimates, and thus the MRR metric decreases. By contrast, the KL-divergence (i.e. loss function) correctly captures the improvement in assessing the relevance of the ground-truth positive. We note that in sparsely annotated datasets like MS MARCO, the "1-hot" ground-truth annotations (right) are very often incorrect among the top ranks, and some of the candidates ranked more highly than the ground truth (e.g. Candidates 3 and 4 in the figure) may actually be relevant, which would render the MRR metric spurious; Qu et al., 2021 estimate that about 70% of the top 5 candidates retrieved by a top-performing dense retrieval model that are not labeled as positive are in fact relevant.

| | Test: | DCTR Head | | | RAW Head | | |
| Model | | MRR@10 | nDCG@10 | Recall@10 | MRR@10 | nDCG@10 | Recall@10 |
|---|---|---|---|---|---|---|---|
| TAS-B | | 0.278 | 0.139 | 0.130 | 0.339 | 0.188 | 0.113 |
| CODER(TAS-B) | | 0.279 | 0.140 | 0.130 | 0.338 | 0.191 | 0.115 |
| CODER(TAS-B) + geom. smooth labels | | 0.285 | 0.143 | **0.134** | 0.344 | **0.195** | **0.116** |
| CODER(TAS-B) + rNN smooth labels | | **0.288** | **0.144** | **0.134** | **0.347** | **0.195** | **0.116** |
| **CODER(TAS-B) + mixed rNN/geom. smooth lab.** | | 0.284 | 0.142 | 0.132 | 0.342 | 0.192 | 0.115 |
| CoCondenser | | 0.242 | 0.114 | 0.105 | 0.293 | 0.157 | 0.092 |
| CODER(CoCondenser) | | 0.251 | 0.117 | 0.107 | 0.306 | 0.161 | 0.093 |
| **CODER(CoCondenser) + mixed rNN/geom. smooth lab.** | | 0.250 | 0.117 | 0.107 | 0.304 | 0.162 | 0.094 |

Table 7: Performance of models trained on MS MARCO but zeroshot-evaluated on TripClick Test. **Bold**: overall best, Underline: best in model class.

| Val: | DCTR Head | | | RAW Head | | |
|---|---|---|---|---|---|---|
| **Model** | **MRR@10** | **nDCG@10** | **Recall@10** | **MRR@10** | **nDCG@10** | **Recall@10** |
| TAS-B | 0.299 | 0.145 | 0.136 | 0.355 | 0.200 | 0.118 |
| CODER(TAS-B) | **0.300** | 0.146 | 0.140 | 0.353 | 0.203 | 0.121 |
| CODER(TAS-B) 
 + geom. smooth labels | 0.297 | **0.147** | 0.140 | 0.355 | 0.204 | 0.121 |
| CODER(TAS-B) 
 + rNN smooth labels | **0.300** | **0.147** | **0.141** | **0.357** | **0.205** | **0.122** |
| **CODER(TAS-B)** 
 **+ mixed rNN/geom. smooth lab.** | 0.299 | **0.147** | **0.141** | 0.355 | 0.204 | **0.122** |
| CoCondenser | 0.247 | 0.115 | 0.105 | 0.308 | 0.167 | 0.097 |
| CODER(CoCondenser) | 0.254 | 0.120 | 0.111 | 0.314 | 0.173 | 0.101 |
| **CODER(CoCondenser)** 
 **+ mixed rNN/geom. smooth lab.** | 0.254 | 0.118 | 0.109 | 0.311 | 0.169 | 0.098 |

Table 8: Performance of models trained on MS MARCO but zeroshot-evaluated on TripClick Val. **Bold**: overall best, Underline: best in model class.

| | Test RAW Torso | | | Val RAW Torso | | |
|---|---|---|---|---|---|---|
| **Model** | **MRR@10** | **nDCG@10** | **Recall@10** | **MRR@10** | **nDCG@10** | **Recall@10** |
| RepBERT | 0.338 | 0.247 | 0.309 | 0.398 | 0.288 | 0.342 |
| CODER(RepBERT) | 0.390 | 0.276 | 0.330 | 0.426 | 0.310 | 0.354 |
| CODER(RepBERT) hyperparam. | 0.389 | 0.277 | 0.331 | **0.428** | 0.310 | 0.354 |
| **CODER(RepBERT)** 
 **+ mixed rNN/geom. smooth label.** | **0.391** | **0.282** | **0.340** | 0.421 | **0.312** | **0.367** |

Table 9: Label smoothing applied to CODER(RepBERT) trained on TripClick HEAD ∪ TORSO Train, validated on HEAD Val; evaluation on TORSO Test and Val.

### A.5.1 Score normalization

In standard contrastive learning, including when using a KL-divergence loss, as in CODER (Zerveas et al., 2022), there is a very stark difference between the probability of the handful of ground-truth documents and the zero probability of the negatives in the target (ground-truth) distribution.

In evidence-based label smoothing, we are using the continuous similarity scores of candidates with respect to the ground-truth positive document(s) as soft labels for training, which means that that there is a reduced contrast between the highest and smallest score values. Additionally, the output values of the model's similarity estimate reside within an arbitrary value range, determined primarily by the model's weights, and for the same rank, there is a large variance of values between queries (Fig. 12). This means that after passing through a softmax, which is highly non-linear, the target score distribution will be either concentrated or diffuse, depending on the range of score values for each particular query. Normalizing values into the same range will facilitate learning consistent relevance estimates. Furthermore, given a single query, we wish that target scores rapidly decrease as the rank increases (Fig. 13).

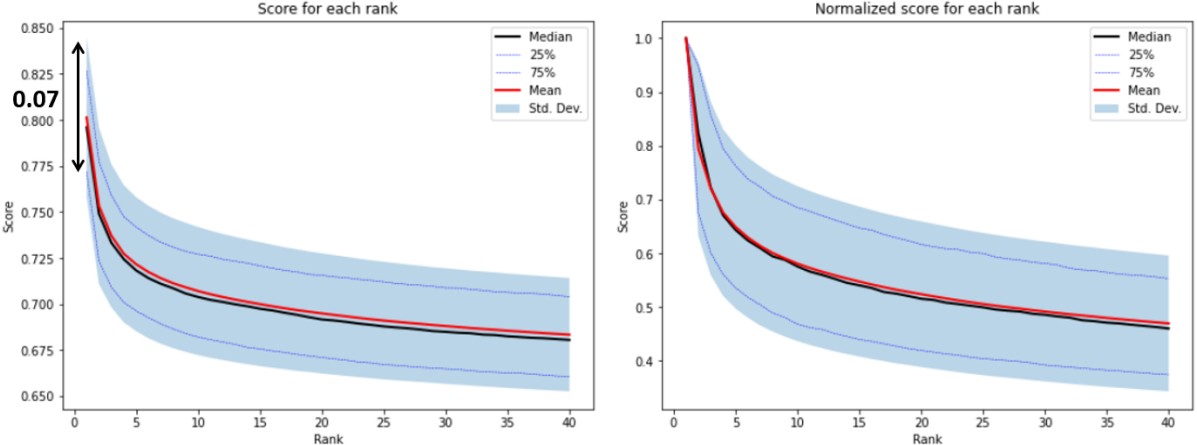

Figure 12: Similarity scores per rank across a large number of queries.

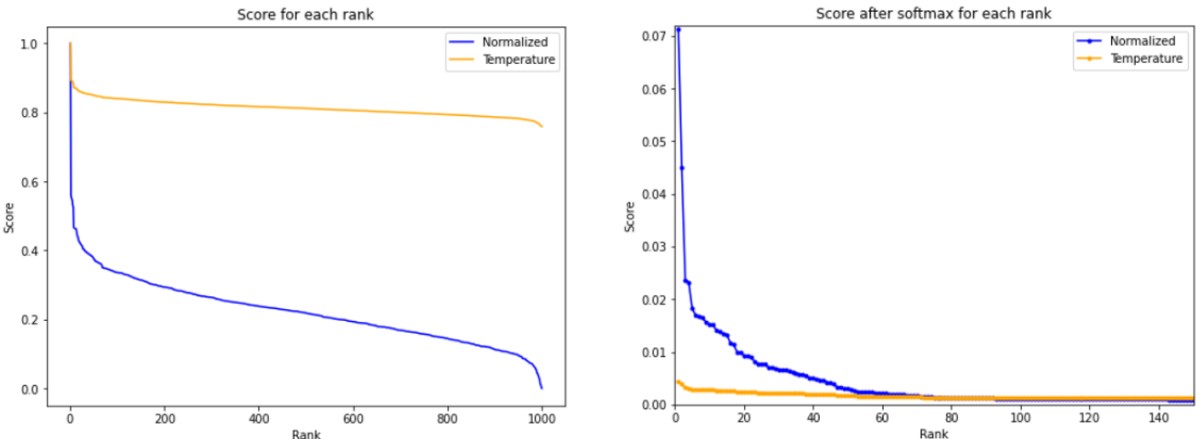

Figure 13: Similarity scores of the top 1000 candidates for a single query, sorted in descending order. Since they are used as training labels, to avoid very diffuse estimated score distributions, we need to ensure that there is a large contrast between the top and bottom candidates and that probability (i.e. values after the scores pass through a softmax) abruptly decreases after the first few ranks. We achieve this through appropriate normalization - here, max-min (blue) instead of dividing by max (orange).

Therefore, to facilitate learning, we wish to ensure that (a) there is large enough contrast between the first and last ranks, and (b) this is true for all queries. We can achieve this by applying a normalizing function $f_n$, such as max-min, on the vector $\mathbf{s} \in \mathbb{R}^N$ of candidate scores for a single query:

$$f_n(\mathbf{s}) = \frac{\mathbf{s} - \min(\mathbf{s})}{\max(\mathbf{s}) - \min(\mathbf{s})} \tag{13}$$

or the following, which is based on the standard deviation $\sigma$ across $N$ candidate scores for a single query:

$$f_n(\mathbf{s}) = \frac{\mathbf{s} - \min(\mathbf{s})}{\sigma} \quad , \tag{14}$$

where $\sigma = \sqrt{\dfrac{\sum_i \left( s_i - \sum_j s_j/N \right)^2}{N}}$.