# OpenReview forum: "Enhancing the Ranking Context of Dense Retrieval through Reciprocal Nearest Neighbors"
_EMNLP/2023/Conference — EMNLP 2023 Main_

### Official Review · Reviewer_GibY · 2023-08-05

**Soundness:** 3

**Excitement:**

4: Strong: This paper deepens the understanding of some phenomenon or lowers the barriers to an existing research direction.

**Paper Topic And Main Contributions:**

The authors proposed an improved similarity metric based on reciprocal nearest neighbors, which can also be used independently to rerank candidates in post-processing.

**Questions For The Authors:**

Can the authors provide more reasons why grouch-truths that are mostly single positive-labeled as in MS MARCO can be ill-suited?

**Reasons To Accept:**

Various and detailed analysis results have shown the soundness of their methods.

The proposed methods are novel and practical.

**Reasons To Reject:**

The authors could have compared with some contrastive learning loss functions, models, etc.

**Reproducibility:**

3: Could reproduce the results with some difficulty. The settings of parameters are underspecified or subjectively determined; the training/evaluation data are not widely available.

**Reviewer Confidence:**

4: Quite sure. I tried to check the important points carefully. It's unlikely, though conceivable, that I missed something that should affect my ratings.

**Typos Grammar Style And Presentation Improvements:**

It could be nicer if the authors define concepts such as geometric similarity s and s_J directly in Section 3.1 instead of the Appendix.

---

> ### Author Rebuttal · Authors · 2023-08-28
>
> We sincerely thank the reviewer for their comments and suggestions, which we will incorporate into the revised paper.
>
> Given the extra page available post-review, we will move definitions from the Appendix to the Methods section, and **we will include a discussion of contrastive learning in a dedicated section of the revised paper**, which we agree is important, as it is the leading training paradigm of dense retrieval methods.
>
> Here, we note the following: The list-wise setting of the CODER framework provides a natural setting for evaluating evidence-based label smoothing, as it jointly scores a large number of candidate documents for which we compute soft relevance labels. However, CODER itself is just a fine-tuning framework and our training starting point is always a model already trained to peak performance through contrastive learning (e.g. RepBERT, TAS-B or CoCondenser). Further training of these models through their original contrastive objectives (including a small set of hard negatives and numerous random documents) leads to deterioration of performance, as already demonstrated by the CODER paper (Zerveas et al, 2022); only a large enough set of pre-retrieved candidate documents within a list-wise loss leads to improved performance through further training/fine-tuning. In our work, we additionally incorporate soft labels when fine-tuning. Therefore, **as we start our training from a converged model trained through contrastive learning, in a sense we already have a reference point for comparison**, e.g., the original CoCondenser on TREC DL 2020 improves from 0.618 to 0.646 nDCG@10 through our method.
>
> Nevertheless, an interesting question here is whether the benefits of evidence-based label smoothing can also be observed _independently_ of using CODER, i.e., _when applied directly to contrastive learning from scratch_, not when fine-tuning with a list-wise objective. We strongly believe so, because false negatives have been shown to be especially deleterious for contrastive learning (e.g.  Qu et al., 2021), and our label smoothing places a larger-than-zero target relevance on false negatives, which would reduce the penalty they incur within the contrastive loss function. Furthermore, the contrastive InfoNCE/Negative Log-Likelihood loss used in most methods when training dense retrievers is _even more susceptible_ to the effect of false negatives than the KL-divergence used in CODER (see discussion Appendix A.5 of Zerveas et al., 2022). Of course, dedicated experiments are required to conclusively demonstrate and quantify this effect.
>
>
> ## Reviewer question: reasons why ground truths that are mostly single positive-labeled as in MS MARCO can be ill-suited
>
> Despite their importance for Information Retrieval research, the process in which large-scale datasets such as MS MARCO are compiled incorporates significant structural biases, many of which are largely inevitable. For every given query, we can naturally assume that there are several relevant documents on the Word Wide Web; yet, only a miniscule subset of those will survive in these datasets - in MS MARCO, almost always just a single one.
>
> First, data is gathered from click logs of a _deployed_ commercial search engine; for a given query, documents that have received the most clicks will be selected and referred to human judges for further confirmation. Thus, the users, and consequently judges, **will only see documents that are most likely to be ranked highly by the deployed search engine** based on its own idiosyncrasies (e.g. a lexical-based model was used by Bing when compiling MS MARCO).
>
> Second, due to the extent of required human effort, each of the hundreds of thousands of queries is typically assigned to a **single human annotator** (with the exception of a miniscule subset, to assess labeling quality). Moreover, each human annotator can only afford to look at **only a handful of top-ranked documents/passages for a given query**, until they identify a document that appears relevant. Specifically, the authors of the MS MARCO paper note (Bajaj et al., 2016): _“As the editors were not required to annotate every passage that were retrieved for the question, this annotation should be considered as incomplete—i.e., there are likely passages in the collection that contain the answer to a question but have not been annotated”._ To create the MS MARCO passage collection and ranking task, which we use in our work, the MS MARCO dataset creators insert an additional bias in the pipeline: after selecting top-ranked documents, they use a separate system to extract the passages most likely to contain an answer, that are then submitted to the human judges. Thus, **for each query, a single human annotator examines at most a couple of passages, extracted by a single system from an even smaller number of documents, retrieved by a single retrieval model.**  It then comes as little surprise that when using a top-performing retrieval model for passage retrieval on MS MARCO, Qu et al. 2021 observe that about **70% of the top 5 candidates not labeled as positive are actually relevant** (i.e. false negatives).
>
> The resulting sparse annotation may be sufficient to _train_ well-performing models; however, when _evaluating_ their ranking effectiveness, models that highly rank the same passage as the one incidentally selected in the dataset through the biased process above will be advantaged compared to models which highly rank passages that might be equally or more relevant, but were not selected in the annotation process. Therefore, **when evaluating on such datasets, models that most effectively learn to imitate the biases present in the selection process** (e.g. the strong lexical overlap heuristic of the Bing search engine pre-2016) **will be advantaged**.
>
> In particular, since such datasets include only a single positive per query, **models which systematically distribute relevance over a larger number of candidates** (like ours do by design, due to label smoothing), **will be _systematically_ underestimated when computing performance metrics such as MRR and nDCG: this is because a larger number of positive but unlabeled documents will receive a high relevance score and will thus more likely displace the ground-truth annotated positive document.** For our models, this happens even when the gound-truth positive document receives a higher relevance score than it did prior to fine-tuning, evidenced by the lower validation loss (see Figure 10 in the Appendix of our paper, and Figure 2).
>
> Consequently, to properly evaluate such methods, one needs datasets with greater annotation depth. The TREC DL 2019, 2020 datasets:
>   1. include **for each query more than 210 passage judgements** on average,
>   2. **several** experienced human annotators examine the same queries,
>   3. judged **passages are retrieved directly from the collection** (not extracted from documents), and
>   4. are **retrieved by a variety of different retrieval systems**
>   5. **relevance ratings are multi-level**, in a scale of 0 (non-relevant) to 3 (perfectly relevant), instead of binary.
>
> Thus, many of the biases of the MS MARCO annotation process are alleviated, with the trade-off that the number of judged queries is much smaller. For this reason, we further evaluate _models trained on MS MARCO_ on the out-of-domain TripClick dataset “zero-shot” (i.e. without additional training). The latter includes 7k evaluation queries annotated with tens of pseudo-relevance labels per query, both in a binary (RAW) and multilevel-level (DCTR) relevance scheme. Thus, _the very same models_ that do not register an improvement when evaluated on MS MARCO dev, show significant improvements on TREC DL 2019, 2020 (Table 3), as well as when zero-shot evaluated on TripClick (Tables 7, 8 in the Appendix). Of course, compared to zero-shot, performance is much better when training on TripClick and evaluating on TripClick (Table 9, Appendix).
>
> Importantly, because of the greater annotation depth of TripClick, the discrepancy between the validation MRR metric and the loss we observe for MS MARCO in Figure 10 (which we hypothesize is due to sparse annotation and false negatives), **is not present** in the corresponding Figure 11 (Appendix) when training on TripClick, where both MRR and loss improve substantially when training with our proposed method (evidence-based label smoothing based on rNN).
>
>
> ### Supplementary comment on MS MARCO dev
>
> Finally, we note that we are not the first to observe puzzling performance behavior when evaluating on MS MARCO dev: in the paper _“MS MARCO Chameleons: Challenging the MS MARCO Leaderboard with Extremely Obstinate Queries”_, CIKM’21, Arabzadeh et al. observe that there are sets of “extremely obstinate queries” on which none of the state-of-the-art neural ranking methods they examined performs well. Furthermore, performance of the models on these sets is uncorrelated with their performance on the rest of the queries. The authors do not find **any special characteristics of these queries that distinguish them from other queries**, and observe that **query reformulation does not help**. While the authors seem to infer that there is something amiss with how state-of-the-art neural ranking methods interpret these queries, and that this set of “obstinate queries” should be therefore used as a special benchmark for ranking performance, we conjecture that the observed effect is explained well through flawed annotation of MS MARCO dev: the existence of a large number of false negatives (which is well documented in literature) means that models cannot highly rank the arbitrarily selected ground-truth positive among other (unlabeled) positives. The large inter-model agreement on which queries are “obstinate”, and the fact that query reformulation is not useful, are consistent with this hypothesis. In our own experiments with all models we considered (TAS-B, RepBERT, CoCondenser), when examining the ranked position of the ground-truth annotated document, we further observed that the ground-truth across all queries in MS MARCO very rarely ranks in middle or bottom ranks (lower than ~50). So it would appear that it is not as much a matter of the models misinterpreting queries (because then the ground-truth would have landed more often in middle and bottom ranks), as it is how low the ground-truth is displaced within the top ranks. Naturally, dedicated experimental work is necessary to verify this hypothesis.

---

### Official Review · Reviewer_BAUY · 2023-08-05

**Soundness:** 4

**Excitement:**

2: Mediocre: This paper makes marginal contributions (vs non-contemporaneous work), so I would rather not see it in the conference.

**Paper Topic And Main Contributions:**

This paper defines a similarity metric based on both inner product and reciprocal nearest neighbors or Jaccard similarity, and shows its effectiveness in inference-time reranking and offline label smoothing of training data.

**Reasons To Accept:**

The proposed similarity definition has marginal improvement on the relevance metrics (MRR, NDCG, etc.) of retrieved results.

**Reasons To Reject:**

The use of reciprocal nearest neighbors may be common practice in research and development. Given the reported scale of improvement in this paper, I would consider it an expected result.

**Reproducibility:**

4: Could mostly reproduce the results, but there may be some variation because of sample variance or minor variations in their interpretation of the protocol or method.

**Reviewer Confidence:**

4: Quite sure. I tried to check the important points carefully. It's unlikely, though conceivable, that I missed something that should affect my ratings.

**Typos Grammar Style And Presentation Improvements:**

The concept of ranking context is used throughout the paper but not formally defined.

Figure 1 in its current form doesn't really help with the intuition in an easy way. It would be better to have multiple subfigures that show the difference between the rankings when (1) different similarity definitions are applied, and (2) new data points are added.

Line 152-157: The claim here is obscure and seems to lack support in references.

---

> ### Author Rebuttal · Authors · 2023-08-28
>
> We thank the reviewer for their comments.
>
> Although we are not aware of the use of reciprocal NN in industry R&D (despite the relevant experience of some of our authorship team members in industry), we cannot exclude that the concept has been used in industry at some point. However, given that we have not been able to find any related references besides the two works on image re-identification cited in our paper, **we still believe that the academic retrieval community would benefit from dissemination of the idea.** We furthermore _quantify the expected performance benefit_ and _provide values for parameters found through optimization_, as well as an _analysis of the effect of key parameters_ such as the context size and the _potential for transferability of these parameters across datasets and models_. We believe that this investigation sets the basis for the use of rNN as a building block in academia and industry alike.
>
>
> More importantly, however, besides simply applying the concept of rNN in a new domain, **the main contribution of this paper consists of devising a new method that allows to effectively leverage it** for improving retrieval performance. Our experiments show that simply using it as part of a similarity metric for post-processing reranking is not the optimal way, as performance improvements are smaller and an extra delay is introduced during inference time. Since this is the only reported use in literature, any related R&D work in industry would most likely follow that approach.
>
> Instead, we propose a completely new method for training retrieval models that utilizes similarity of candidate documents with respect to ground-truth documents  - _independent of rNN_ - to compute soft relevance labels. Using these computed labels within an appropriate list-wise loss allows us to mitigate the problem of false negatives when training dense retrieval models. It is within this new framework where the concept of rNN proves most valuable, and we provide all additional methodological techniques (e.g. normalization, score value boosting, hard candidate cut-off, etc.) that are required for it to work.
>
> Regarding the magnitude of performance improvements, we agree that it appears modest in absolute terms. However, as we discuss in the paper, due to its structural biases (biased selection of a single positive per query), **the MS MARCO dev dataset systematically underestimates the performance of methods that distribute relevance scores over multiple candidates** rather than concentrating scores on a couple candidates. When evaluated in a suitable setting, such as the TREC DL 2019, 2020 and TripClick datasets, our proposed method offers a _performance boost of about +0.010 nDCG@10, which is considered a significant improvement in IR literature_.
>
> Importantly, **this performance benefit comes “for free”**: there are no trade-offs, and given an existing model ready for deployment, one would only need to compute soft labels once (offline, fully paralllizable on CPUs), and then fine-tune the model for a few hours on a single GPU.
>
>
> We thank the reviewer for their suggestions **regarding presentation**:
>
> given the additionally available space post-review, we will split Figure 1 into a sequence of subfigures.
> we will explicitly define _ranking context_ early on in the paper. As ranking context we mean the set of candidate documents that are jointly taken into consideration when evaluated with respect to their similarity to the same query.
>
> Lines 152-157: These lines correspond to our comment that “the existence of a model that is more powerful than the retrieval model we wish to train is a very restrictive constraint, and cannot be taken for granted in many practical settings”. By this we simply mean that when training a retrieval model, it is not guaranteed that there will always be an even more powerful model to use for pseudo-labeling or distillation. We consider this to be a modest and intuitive claim, but we can certainly improve our wording to facilitate clarity.

---

### Official Review · Reviewer_LNyP · 2023-08-06

**Soundness:** 4

**Excitement:**

4: Strong: This paper deepens the understanding of some phenomenon or lowers the barriers to an existing research direction.

**Paper Topic And Main Contributions:**

This paper proposes an evidence-based label smoothing method, which mitigates the problem of false negatives by leveraging the similarity of candidate documents within the ranking context of a query to the annotated ground truth in order to compute soft relevance labels. Experiments demonstrate the effectiveness of the proposed methods.


**Reasons To Accept:**

This paper innovatively utilizes the rNN method, essentially mining additional relationship information between queries and documents, to obtain more effective hard negative samples and improve training efficiency.

The paper addresses the efficiency of the proposed methods, which is a commendable attempt in academic research.

**Reasons To Reject:**

I believe the method proposed in this paper, using rNN to improve the estimation of semantic similarity between query and document embeddings, is indeed novel. However, the challenges mentioned at the beginning of the paper, such as the sparsity of relevance annotations, have long been present in the field of Information Retrieval even before the advent of Dense Retrieval. There have been many similar discussions with similar ideas but different methods. I believe these works should also be cited and discussed, including:

Qingyao Ai, Keping Bi, Jiafeng Guo, and W. Bruce Croft. 2018. Learning a Deep Listwise Context Model for Ranking Refinement. In The 41st International ACM SIGIR Conference on Research & Development in Information Retrieval (SIGIR '18). Association for Computing Machinery, New York, NY, USA, 135–144. https://doi.org/10.1145/3209978.3209985

Diaz F. Regularizing query-based retrieval scores[J]. Information Retrieval, 2007, 10: 531-562.

**Reproducibility:**

3: Could reproduce the results with some difficulty. The settings of parameters are underspecified or subjectively determined; the training/evaluation data are not widely available.

**Reviewer Confidence:**

4: Quite sure. I tried to check the important points carefully. It's unlikely, though conceivable, that I missed something that should affect my ratings.

---

> ### Author Rebuttal · Authors · 2023-08-28
>
> We are grateful to the reviewer for their comments and suggestions, especially with respect to related work in IR before the advent of Dense Retrieval.
>
> We entirely agree that a discussion of this work should be included, and funnily enough, in an earlier draft of our paper, we had a small section exactly on earlier Learning-to-Rank work demonstrating the importance of ranking context, including the Ai et al. 2018 paper. This was unfortunately cut due to aggressive downsizing to meet the 8 pages limit. Thankfully, the extra page available post-review will allow us to restore this section, as well as include the interesting Diaz 2007 paper, which, besides being a different method addressing the same phenomenon from a different angle, provides important additional background and justification for our work.

---

### Meta-Review · Area_Chair_eqv2 · 2023-09-20

**Recommendation:** 5

**Metareview:**

The paper introduces evidence-based label smoothing, a computationally efficient method that prevents assigning high relevance to false negatives by leveraging the similarity of candidate documents within the ranking context of a query to the annotated ground truth to compute soft relevance labels. Extensive experimentation on two large-scale ad hoc text retrieval datasets demonstrates that the method improves the ranking effectiveness of dense retrieval models. Overall a nice paper. Well well-written and well-supported by reproducible experimentation.

---

### Decision · Program_Chairs · 2023-10-07

**Decision:**

Accept-Main

**Comment:**

The paper introduces evidence-based label smoothing, a computationally efficient method that prevents assigning high relevance to false negatives by leveraging the similarity of candidate documents within the ranking context of a query to the annotated ground truth to compute soft relevance labels. Extensive experimentation on two large-scale ad hoc text retrieval datasets demonstrates that the method improves the ranking effectiveness of dense retrieval models. Overall a nice paper. Well well-written and well-supported by reproducible experimentation.